# Selecting police super-recognisers

**James D. Dunn** [1], **Alice Towler** [1,2], **Richard I. Kemp**[1], **David White** [1]*

**1** School of Psychology, UNSW Sydney, Sydney, Australia, **2** School of Psychology, University of Queensland, Brisbane, Australia

* david.white@unsw.edu.au

## Abstract

People vary in their ability to recognise faces. These individual differences are consistent over time, heritable and associated with brain anatomy. This implies that face identity processing can be improved in applied settings by selecting high performers–'super-recognisers' (SRs)–but these selection processes are rarely available for scientific scrutiny. Here we report an 'end-to-end' selection process used to establish an SR 'unit' in a large police force. Australian police officers (n = 1600) completed 3 standardised face identification tests and we recruited 38 SRs from this cohort to complete 10 follow-up tests. As a group, SRs were 20% better than controls in lab-based tests of face memory and matching, and equalled or surpassed accuracy of forensic specialists that currently perform face identification tasks for police. Individually, SR accuracy was variable but this problem was mitigated by adopting strict selection criteria. SRs' superior abilities transferred only partially to body identity decisions where the face was not visible, and they were no better than controls at deciding which visual scene that faces had initially been encountered in. Notwithstanding these important qualifications, we conclude that super-recognisers are an effective solution to improving face identity processing in applied settings.

## 1 Introduction

Identifying faces underpins everyday social interactions, and many important applied identity processing tasks. Staff working in forensic and security settings often perform face identity processing tasks in large volumes. For example, passport officers compare faces of travellers to their passports, CCTV operators monitor surveillance for persons of interest, and forensic facial examiners prepare image comparison reports for use by courts and in criminal investigations. Although facial recognition technology is increasingly used to reduce the need for human processing in many of these applied tasks, this technology requires human oversight by people with sufficient skill to detect errors [1].

Human errors in these tasks can have profound consequences for individuals and for society, for example wrongful convictions or terrorist attacks. But human errors are also surprisingly common, with experimental studies asking participants to decide if two images show the same person reporting error rates between 30–40% for challenging tasks [2,3] to 20% when conditions are optimised for high accuracy [4]. Similar error rates are reported in studies of professional groups that perform face image comparison tasks in their daily work: groups of

**Funding:** This research was supported by Australian Research Council funding to White (DP190100957; FT200100353) and private funding from the New South Wales Police Force. The funders had no role in study design, data collection and analysis, decision to publish, or preparation of the manuscript.

**Competing interests:** The authors have declared that no competing interests exist.

bank tellers, security, passport and police officers all achieve equivalent accuracy to groups of university students [5].

A promising solution to this problem has emerged from the science of face perception. It is now firmly established that face identity processing ability varies between individuals to a surprisingly large extent. This ability ranges from people with prosopagnosia, who are unable to recognise even their closest friends and family members [6], to super-recognisers who consistently attain the highest levels of accuracy on tests of face identification ability [7]. These extremes form endpoints of a normally distributed cognitive trait that varies dimensionally in the broader population [8].

Several published tests are designed to measure face identity processing ability across the ability spectrum (see [8] for a recent review, see also [9]). The most widely used is the Cambridge Face Memory Test which measures the ability to learn and remember unfamiliar faces [10]. Research has established that a person's CFMT score is stable over time [11], that there is a substantial genetic contribution to performance [11–13], and that variation in CFMT scores are associated with measures of brain structure and physiology (e.g. [14,15]). Similar levels of stability in individuals' performance are found in face matching tests that do not involve memory but instead deciding whether images shown at the same time are of the same person [16]. There is also high correlation between tests of face identification with diverse task demands, pointing to a common, trait-like ability that generalises across identity processing tasks (e.g. [17,18]).

Practical outcomes have stemmed from this research. Police, government and private organisations have used face identification tests to select staff for specialist operational roles [19]. The idea of using tests of face identity processing to select teams of face identification specialists was first explored by the London Metropolitan Police [20,21], and subsequently by the Australian Passport Office [22], Queensland [23] and Berlin police forces [24,25]. Some of the groups resulting from these selection processes have been formally tested and show higher accuracy in small-scale studies. For example, London Metropolitan police SRs outperform control participants by 10 percentage points on average in [20] ($n = 10$) and 17 percentage points in [21] ($n = 4$).

However, it is typically unclear exactly how members of these super-recogniser teams were initially selected, and details of their professional experience and training are unknown (but see 25). Moreover, the full testing process leading to their recruitment has not yet been formally reported, and so gains in accuracy that are attainable through staff selection alone have not been quantified. One study estimated the gains of selecting a small cohort of undergraduate students ([26]; $n = 114$), and found that selecting the top 10 performers could produce accuracy gains of 8%. Comparable accuracy differences between super-recognisers selected from the general public and standard comparison groups have been found in other studies, ranging from 8% to 20% (e.g. [27–29]). Other studies have compared super-recognisers selected from the general population to existing professional cohorts of forensic facial examiners and found equivalent accuracy [3,23,30]. This work points to the potential for super-recognisers to improve accuracy in forensic settings. Importantly however, these studies do not evaluate end-to-end procedures for selecting super-recognisers in organisations.

Here we provide a detailed report of the processes used to select super-recognisers in the New South Wales Police Force, a large Australian organisation with approximately 15,000 police officers and 4,000 civilian staff. Our main aim was to provide a data-driven roadmap for selecting super-recognisers on the basis of face identity processing tests, quantifying the potential benefit of adopting this approach for similar organisations.

To do this, we first used online tests to identify individuals who met specific performance criteria on three standardised tests of unfamiliar face memory and face matching ability.

Because there is no universally agreed criterion for selecting super-recognisers (see [9], c.f. [31]) we applied two alternative selection criteria and assessed the impact of these criteria on group performance in follow-up tests in our research lab. Follow-up testing included challenging face matching tests used in prior work, to benchmark their accuracy against high performing professional groups of forensic facial examiners [3,22,23,32]. Forensic facial examiners are the most commonly used face identification experts in forensic and security settings, providing a useful standard for assessing super-recognisers potential value. We also designed bespoke tests that captured a wider range of task demands encountered in police work and compared super-recogniser performance on these to standard participant groups.

To pre-empt our results, we find comparable accuracy between super-recognisers and existing professional experts on a range of unfamiliar face matching tasks that are representative of police work. Indeed, super-recognisers were far more accurate than these comparison groups when limited to study the faces to just two seconds. Face inversion effects in super-recognisers and facial examiners also point to qualitative differences in the nature of their face identity processing.

When comparing to standard participant groups of university students, we find that super-recognisers had consistently higher accuracy on challenging applied tasks (Method & Results section 1). Summary analysis shows that this superiority was notably larger when super-recognisers were selected by a strict versus weak selection criteria (see Method & Results section 3). More detailed cognitive testing shows that super-recognisers also have a heightened ability to perform perceptual comparison tasks on non-face stimuli, but these abilities appear to be graded by the extent to which the perceptual information is person-related (Face > Body > Houses; see Method & Results section 2). We also show an important limitation of super-recognisers' face recognition abilities: While they were far better at recognising unfamiliar faces, they were no better at remembering the context in which they initially encountered the face (Method & Results section 2).

## 2 Method and results

The data that support the findings of this study are openly available on the Open Science Framework (https://doi.org/10.17605/OSF.IO/BJDWK).

Super-recognisers (n = 38) were selected from a cohort of 1600 New South Wales Police Force employees that completed 3 screening tests on their workplace computers. Screening tests were three standardised tests of face matching and memory administered online: the Glasgow Face Matching Test (GFMT; [4]); the Cambridge Face Memory Test long form (CFMT+; [7]), and the UNSW Face Test [33]. Average performance of the 1600 police officers was consistent with normative data from standard participant groups, except they exceeded norms on the GFMT (see Table 1 in Methods).

We applied two selection criteria to scores on these screening tests. This produced two overlapping groups of participants that we invited for further cognitive testing in our lab. "Strict SR" criteria participants achieved scores that exceeded 1.7 standard deviations above

**Table 1. Comparison of NSW Police Force staff to normative data on 3 screening tests.** See screening tests section above for sources of normative data.

|  | Normative | | NSW Police | | |
|---|---|---|---|---|---|
|  | *Mean* | *SD* | *Mean* | *SD* | *N* |
| CFMT+ | 69.6 | 11.5 | 70.0 | 11.7 | 2019 |
| GFMT | 81.3 | 9.7 | 89.5 | 8.2 | 2465 |
| UNSW | 58.9 | 5.8 | 59.7 | 8.0 | 1742 |

the published normative mean on *all* three screening tests (SR-Strict, *n* = 15 identified; *n* = 11 participated). "Weak SR" criteria participants, which included the participants in the strict criteria group, had an *average score* across the three screening tests that exceeded 1.7 standard deviations above the published normative mean (SR-Weak, *n* = 54 identified; *n* = 38 participated). Full details of the method for selecting super-recognisers are provided in Methods (see Participants section).

Follow-up cognitive testing in our lab had four main aims. First, it was important to ensure that individuals selected as super-recognisers could demonstrate sustained levels of high accuracy on a series of challenging face identification tasks with varying task demands many weeks after the initial screening tests. Second, it enabled us to compare the face identification performance of groups selected by strict versus weak criteria, to quantify the benefit of selecting face identification specialists in applied settings (summary analysis comparing these groups across all tests is given in the final Method and Results section). Third, we tested for quantitative and qualitative differences in super-recogniser abilities relative to professional experts in forensic face identification who are routinely employed in police forces. Fourth, we tested the boundary conditions of super-recognisers' abilities on tasks that do not involve face processing but are nevertheless important in police work.

## 2.1. Comparing police super-recognisers to facial examiners on challenging face identification tests in the lab verifies their superior ability

Next, we tested super-recognisers' ability to perceptually match face identity on challenging tasks routinely performed in forensic settings. These tests were challenging because images of the same face used in matching test items were taken at different times, often years apart, in different environmental conditions and using different cameras. For items where images were of different faces, nonmatching 'foil' images were selected as similar to target faces using facial recognition technology. In addition to making the task more challenging, selecting images in this way reflects many modern applied tasks in forensic police work where officers often use facial recognition technology to assist with investigations (e.g., [22]). Because all of these tests have been used in prior work to test facial forensic examiners, it enabled us to benchmark accuracy of super-recognisers to an established industry standard. Examples of test items from the facial recognition candidate list test [22] and the selfie-to-passport test and shown in Fig 1 and further details of these tests are provided in the Methods.

**2.1.1 Facial recognition candidate list test.** As shown in Fig 1, both groups of super-recognisers outperformed Controls, and were as accurate as Facial Examiners on this task (SR-Weak vs. Control: $t(77.4) = 6.77$, $p < .001$, Cohen's $d = 1.49$; SR-Strict vs. Control: $t(33.8) = 10.60$, $p < .001$, Cohen's $d = 2.33$; SR-Weak vs. Examiners: $t(7.4) = 0.55$, $p = .597$, Cohen's $d = 0.27$; SR-Strict vs. Examiners: $t(6.9) = 0.64$, $p = .541$, Cohen's $d = 0.38$).

We also found that the SR-Strict group ($M = 73.4\%$, $SD = 5.9\%$) was numerically more accurate than the SR-Weak group ($M = 65.4\%$, $SD = 13.1\%$) (Mean difference = 8.0%, Cohen's $d = 0.67$). The fact that the SR-strict group were a subset of the SR-weak group makes inferential tests problematic, but the difference in accuracy between these groups is important in applied settings to guide policy decisions. Therefore, in each section that follows we present effect size in terms of percentage points and test the consistency of this difference across all tests reported below in the final section of Methods and Results (see *Summary performance profiles show benefits of stricter criteria when selecting groups of super-recognisers*).

**2.1.2 Selfie-to-passport test.** We found equivalent accuracy in the SR-Strict group and Facial Examiners but the SR-Weak group was significantly less accurate than Facial Examiners (SR-Weak vs. Examiners: $t(35.8) = -3.55$, $p = .001$, Cohen's $d = -0.63$; SR-Strict vs. Examiners: $t$

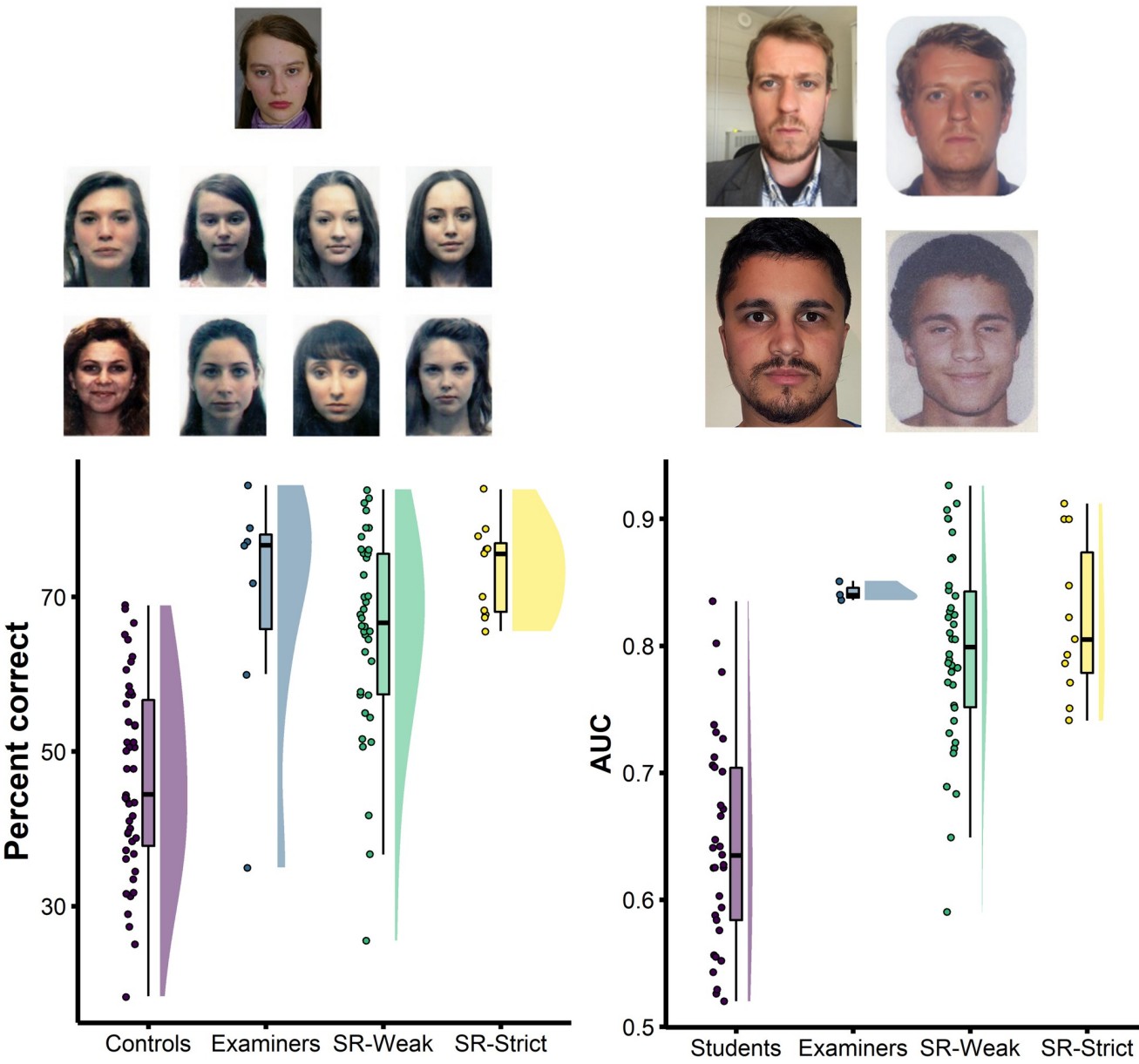

**Fig 1. Super-recognisers' face matching accuracy benchmarked against student controls and forensic examiners on two challenging tests of face matching.** In the facial recognition candidate list test (left), participants decided if the face pictured at the top matched any in the array ('candidate list') below. In the selfie-to-passport test (right), participants decided if the two images showed the same person or two different people. Super-recogniser groups were equivalent in accuracy compared to examiners on both tasks. Despite this, average performance was substantially less than 100% and no individual attained perfect accuracy.

(11.0) = 1.14, $p$ = .280, Cohen's $d$ = 0.39). Both the SR-Weak and SR-Strict groups were more accurate than Controls (SR-Weak vs. Controls: $t(65.9)$ = 8.08, $p < .001$, Cohen's $d$ = 1.93; SR-Strict vs. Controls: $t(22.9)$ = 7.59, $p < .001$, Cohen's $d$ = 2.29). There was a small numerical

advantage for the SR-Strict ($M = 0.82$, $SD = 0.06$) over the SR-Weak group ($M = 0.80$, $SD = 0.08$; Mean difference = 0.03, Cohen's $d = 0.34$).

**2.1.3 Qualitative differences between super-recognisers and facial examiners in face matching.** Prior work has shown qualitative differences in the way that facial examiners perform face matching tasks relative to student comparison groups. In the Expertise in Facial Comparison Test (EFCT, [32]; see Fig 2), examiners' superiority over controls was only evident when given sufficient time to study images, and not when asked to make quick judgments (see also [23]). Facial examiners also showed less impairment as a result of viewing the faces upside down, suggesting more feature-based comparison strategy to the task relative to typical viewers (see also [34]). However, unlike facial examiners–whose face identification abilities are the product of training and experience (see [5,35])–super-recognisers' abilities stem from natural variation in individual ability, and so we might expect differences in their perceptual processing. Here we compared these qualitative patterns in super-recognisers to facial examiners using the EFCT (see Fig 2, top; see Methods for further details).

We compared qualitative patterns of performance on the EFCT with two separate 2-way ANOVAs described below. First, comparing accuracy on upright faces for groups in 2s Vs 30s study duration. Second, comparing the size of Face Inversion Effects in 2s Vs 30s study duration. We also conducted a 4 x 2 x 2 mixed factors ANOVA with Group as the between-subjects factor and Study Duration (2s, 30s) and Face Orientation (upright, inverted) as the within-subjects factors. The full three-factor analysis is reported in S1 Appendix.

*Study duration*. A 4 x 2 mixed ANOVA revealed a significant interaction between Group and Study Duration ($F(3,104) = 7.55$, $p < .001$, $\eta_p^2 = .18$), indicating that the relative accuracy of groups varied as a function of study duration. For the 2 seconds condition, follow-up comparisons show that both SR-Weak and SR-Strict super-recogniser groups were more accurate than Examiners (SR-Weak vs. Examiners: $t(48.3) = 3.91$, $p < .001$, Cohen's $d = 1.02$; SR-Strict vs. Examiners: $t(32.6) = 4.71$, $p < .001$, Cohen's $d = 1.33$). There was a small numerical advantage for the SR-Strict ($M = 0.91$, $SD = 0.04$) over the SR-Weak group ($M = 0.88$, $SD = 0.06$; Mean difference = 0.02, Cohen's $d = 0.37$).

For the 30 seconds condition, SR-Weak group ($M = 0.94$, $SD = 0.04$) and SR-Strict group ($M = 0.95$, $SD = 0.03$) were both statistically equivalent to Examiners (SR-Weak vs. Examiners: $t(57.9) = 0.67$, $p = .506$, Cohen's $d = 0.16$; SR-Strict vs. Examiners: $t(23.9) = 1.79$, $p = .086$, Cohen's $d = 0.58$). Again there was a small numerical advantage for the SR-Strict over SR-Weak group (Mean difference = 0.01, Cohen's $d = 0.37$). Thus, super-recognisers were more accurate than examiners when viewing face images for very short durations (see also [23]).

*Face inversion effect*. Face inversion effects for each participant were calculated by subtracting AUC scores on inverted from upright face conditions, separately for 2 and 30 second. A 4x2 mixed factors ANOVA revealed a significant interaction between Group and Study duration ($F(3,104) = 3.51$, $p = .018$, $\eta_p^2 = .09$; see bottom panel of Fig 2). Follow-up comparisons for the 2 seconds condition showed SR-Weak ($M = 0.18$, $SD = 0.08$) and SR-Strict ($M = 0.19$, $SD = 0.07$) had larger inversion effects than Examiners, but were no different to Student controls (SR-Weak vs. Examiners: $t(62.9) = 2.98$, $p = .004$, Cohen's $d = 0.71$; SR-Strict vs. Examiners: $t(16.0) = 2.4$, $p = .028$, Cohen's $d = 0.93$; SR-Weak vs. Students: $t(63.2) = 0.41$, $p = .684$, Cohen's $d = 0.10$; SR-Strict vs. Students: $t(22.0) = 0.56$, $p = .578$, Cohen's $d = 0.18$).

For the 30 seconds condition, we observed a strikingly different pattern. Face Inversion Effects in SR-Weak ($M = 0.11$, $SD = 0.06$) and SR-Strict ($M = 0.09$, $SD = 0.06$) were no different to Examiners, and were significantly smaller than Student controls (SR-Weak vs. Examiners: $t(62.4) = 0.09$, $p = .930$, Cohen's $d = 0.02$; SR-Strict vs. Examiners: $t(16.1) = 1.07$, $p = .298$, Cohen's $d = 0.41$; SR-Weak vs. Controls: $t(63.4) = 2.39$, $p = .020$, Cohen's $d = 0.58$; SR-Strict vs.

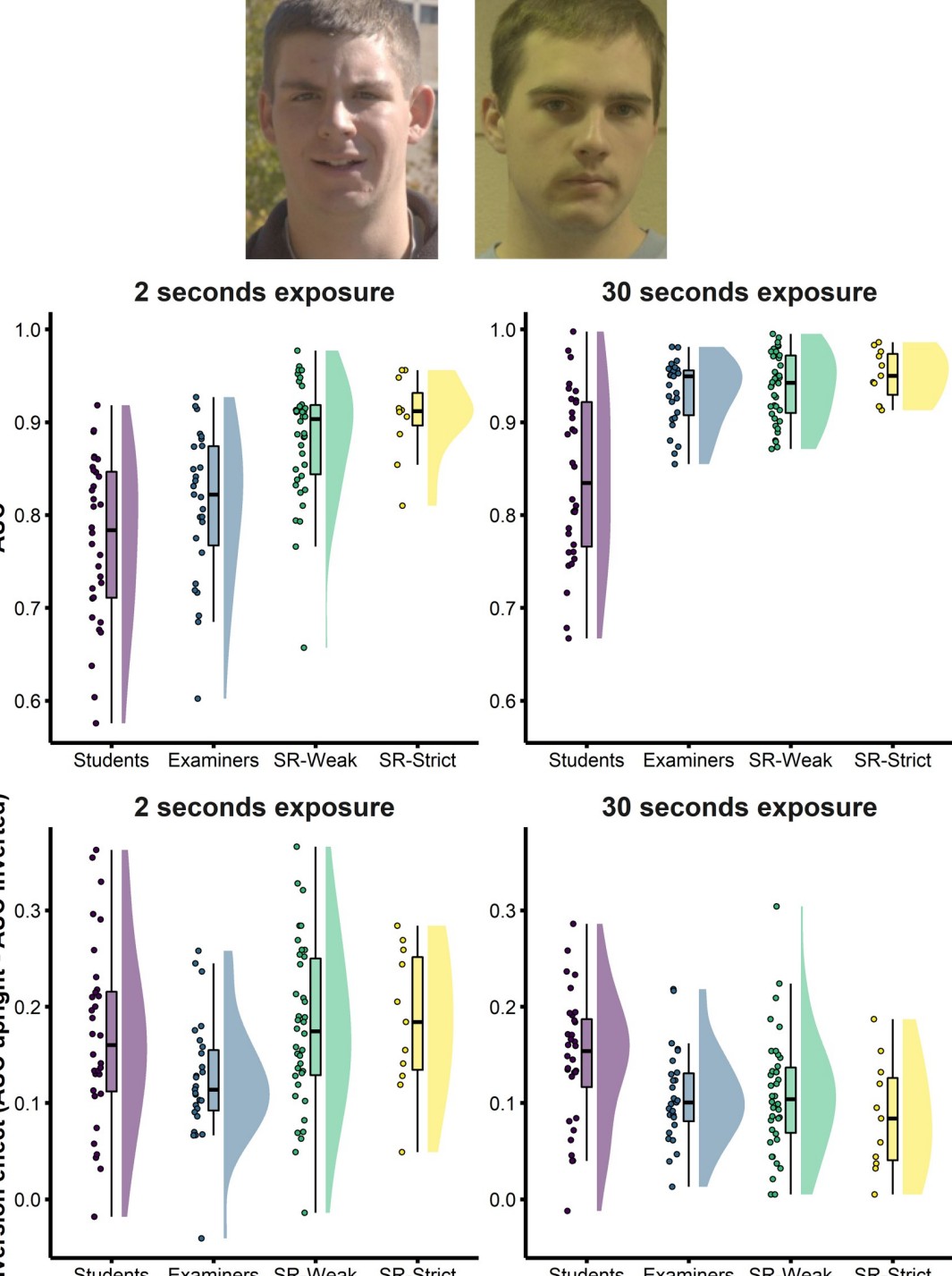

**Fig 2. Super-recogniser performance compared to student controls and facial forensic examiners on the Expertise in Facial Image Comparison Test (EFCT; White, Phillips et al. 2015).** Participants decided whether two images, taken months apart and in different environmental conditions, showed the same person or two different people (top). Super-recognisers were more accurate than examiners when making decisions under a short time limit (2 seconds, top left panel) but not when participants were given longer to study the images (30 seconds, top right). Qualitative differences between groups were also observed in the effect of turning faces upside down (Inversion effect, bottom panels: See text for details).

Controls: $t(21.0) = 2.84$, $p = .010$, Cohen's $d = 0.91$). Together, this pattern of results suggests that super-recognisers are as affected by inversion as control participants with shorter study durations but behave more like examiners at longer durations. This may indicate that super-recognisers are able to adapt their perceptual processing strategies depending on the demands of the task more readily than both examiners and controls.

## 2.2. Establishing the boundaries of super-recognisers' superior abilities

Having established that our super-recogniser groups were comparable to existing face identification experts in perceptual face matching, we asked whether they were also superior in a broader range of tasks they might expected to perform in police work. The range of operational person identification tasks that police officers perform extends beyond the lab-based face identification tests that are typically used to measure ability [7], but the boundaries that define their superior perceptual and cognitive ability are yet to be fully established. For clarity, the following analyses compared only SR-Weak group–i.e. the full sample of SRs–to student controls, but similar patterns were observed when we limited the anlysis to only those SRs meeting the strict selection criteria (see S1 Appendix).

**2.2.1 Matching person identity when the face is not visible.** When matching images from CCTV, face information is often unavailable due to occlusion, head angle or poor image quality. Can super-recognisers also achieve superior accuracy when matching identifying information from the body? Previous work has shown an association between individual differences in face and body identification tasks [36], perhaps pointing to recruitment of a broader person identification system that recruits diverse sources of identity information [37]. We presented participants with two images of people taken at different times in different environmental settings and asked them to decide if they showed the same person or different people using a Likert scale. Images either showed only the internal features of the face (face only) or only information outside of the internal features (body only; see Fig 3 and methods for more details).

Visual inspection of Fig 3 shows that super-recognisers (SR-Weak) outperformed Student Controls in both Face Only and Body Only conditions, but this advantage was largest for Face Only images. A mixed-factor ANOVA confirmed this interaction was significant ($F(1,82) = 12.99$, $p = .001$, $\eta_p^2 = .14$). Simple main effects showed larger differences in accuracy for Face Only trials compared to Body Only trials (Face Only: Mean difference = 23%, $t(81.4) = 7.32$, $p < .001$, Cohen's $d = 1.56$; Body Only: Mean difference = 13%, $t(80.8) = 6.41$, $p < .001$, Cohen's $d = 1.36$). Full details of the ANOVA analysis are provided in S1 Appendix.

**2.2.2 Memory for faces and the context in which they are encountered.** All of the lab-based face identification tasks presented so far involved perceptual matching, not memory. However, in many surveillance and security scenarios, it is important to retain faces in memory. For example, police super-recognisers are often asked to commit large numbers of facial photos of suspects to memory prior to large public events, aiming to recognize them in the crowds [38].

For more spontaneous recognitions, for example when recognising a suspect in CCTV from a previous crime, it is also important for super-recognisers to remember *where they know the face from*. A police officer might for example recognise a person in a CCTV recording, but not realise the person works at their local coffee shop. So, we also tested their ability to link memory of faces to scenes in which they were encountered. Decades of memory research have established a distinction between recognition-based and recollection-based memory (e.g. [39]). Recollection is an intentional and more difficult form of memory processing compared with the automatic recognition memory. This distinction is often referred to as the 'butcher on

# Face and Body Matching Test

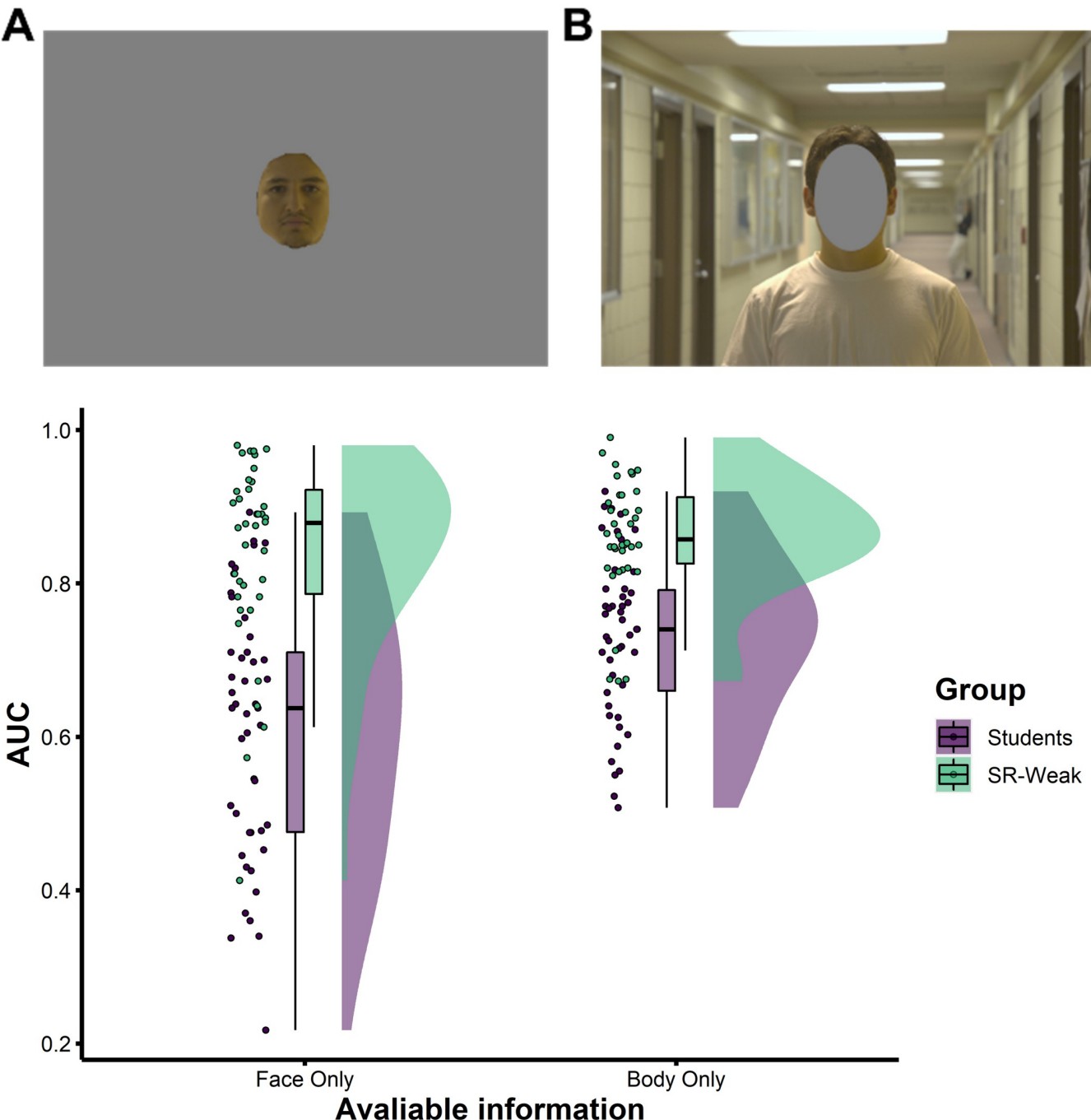

**Fig 3. Super-recognisers' superiority is largest when faces are visible.** Examples of Face Only (A) and Body Only (B) stimuli used in the Face and Body Matching Test. Super-recognisers' superiority is largest when matching identities with the faces visible (Face Only), but they also show an advantage when matching identities using only information in the body (Body Only). Comparison of Student Controls and super-recognisers are shown using the SR-Weak criteria only, but similar patterns emerge when using the SR-Strict criteria (see S1 Appendix).

the bus' phenomenon, because it captures the everyday experience of recognising someone that you know but being unable to recall where you know them from (see [40,41]). Whether super-recognisers are also better able to link face memories to context is therefore of practical importance and can help us understand the person-related memory representations associated with high accuracy in face recognition.

*Face memory from police 'photoboards'*. To test face memory ability, we created an experimental analogue of 'photoboards' used to familiarise super-recognisers with people they would later be expected to recognise in the field. An example of the photoboard test is shown in Fig 4.

# Photoboard recognition memory test

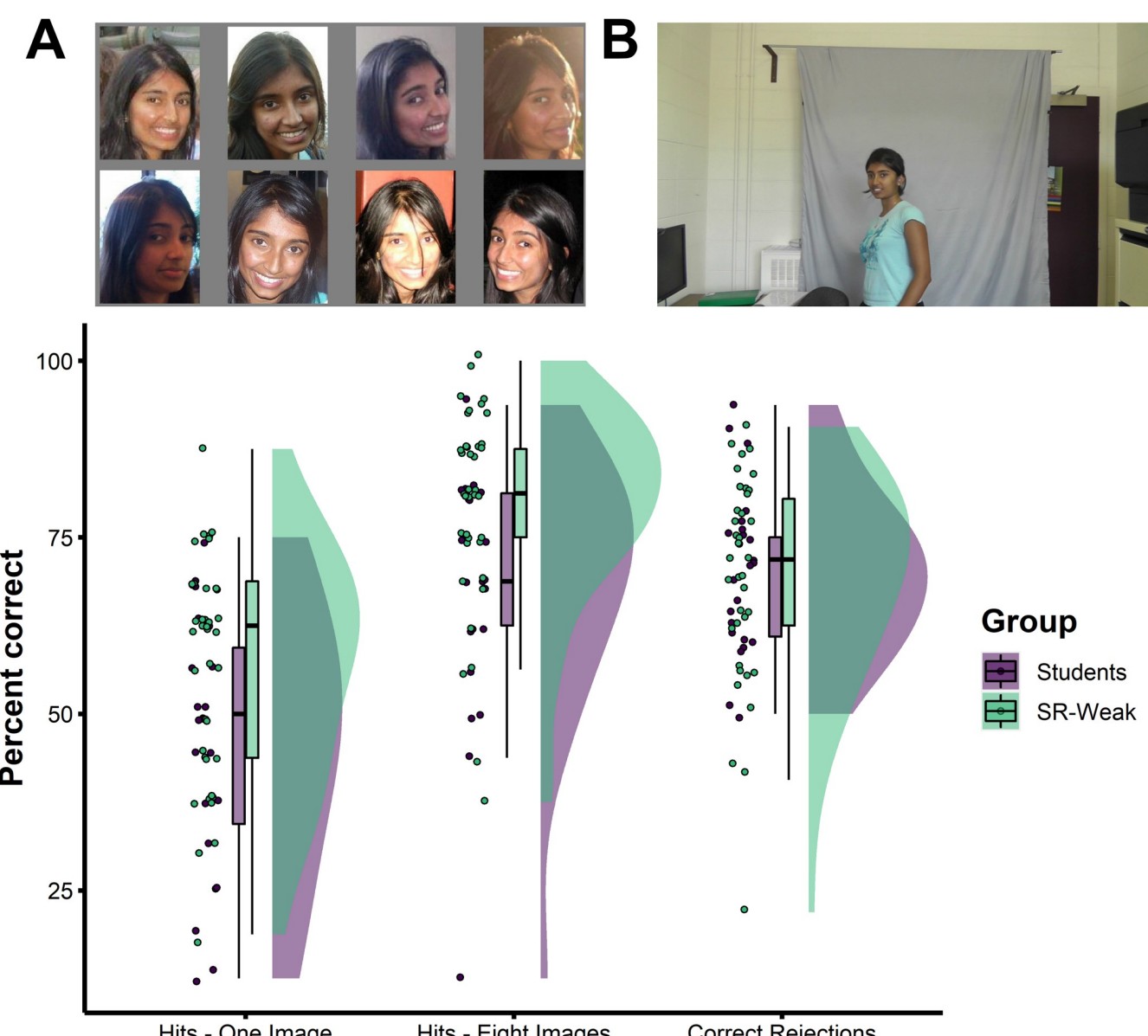

**Fig 4. Super-recognisers show superior face memory in the photoboard recognition memory test.** This test measures participants' ability to learn faces from either one or multiple photographs. Example stimuli from the study phase (A) and test phase (B). Super-recognisers (SR-Weak) were more accurate than Student Controls when recognising faces learnt from one image or eight images but did not show any additional benefit from having access to more images.

In the study phase, participants were shown faces pictured in photoboards containing either 8 images from social media of a single face, or a single image of the face. In the subsequent test phase, they were shown videos of these individuals intermixed with foils that were not shown in the study phase.

Given that super-recognisers sample more information from faces [42] and make identity decisions relatively quickly [23], we predicted that their advantage over controls would be greatest for photoboards containing multiple images of the same face, which would enable them to construct more detailed identity representations in memory. Results did not support this prediction, with super-recognisers (SR-Weak) showing the same advantage over Controls in 1 images and 8 image conditions (main effect of photoboard size: $F(1, 59) = 11.04$, $p = .002$, $\eta_p^2 = .16$; non-sig interaction: $F(1, 59) < 1$, $p = .933$, $\eta_p^2 = .00$; see S1 Appendix for further details).

*'Butcher on the bus'*: *Face-in-place recognition memory test*. Having verified superior face memory in super-recognisers, we then asked whether they also had an advantage when linking memories of faces to the visual scenes in which they were encountered. In this test, participants were asked to remember a series of unfamiliar faces superimposed on scenes. Later, they were shown these faces again mixed with unseen people, both of which were again superimposed on scenes. The studied faces were either shown on the same scenes as during study or on different scenes. They were asked to decide whether the face had been shown in the learning phase (i.e. face recognition accuracy). If they responded that they had seen the face during learning they were also asked whether it had appeared on the same or different scene to the one currently displayed (Face-in-place recognition; see Fig 5).

Results of the face-in-place recognition memory test are shown in Fig 5. For face recognition accuracy, there was a significant main effect of Group, with a higher hit rate in Super-recognisers (SR-Weak: M = 64.3; SD = 16.5) compared to Students (M = 49.6; SD = 14.7; $F(1,56) = 14.05$, $p < .001$, $\eta_p^2 = .20$), and a higher correct rejection rate ($t(35.7) = 2.28$, $p = .028$, Cohen's $d = 0.66$; see S1 Appendix for full ANOVA analysis). Crucially, for accuracy on the subsequent '*did you learn this face in this scene*?' decision, the difference between super-recognisers (M = 63.7; SD = 14.0) and Students (M = 62.9; SD = 16.4) was not significant ($t(36.6) = 0.19$, $p = .854$, Cohen's $d = 0.05$). These results show that while super-recognisers were better at recognising faces seen in the learning phase regardless of whether they were seen in the same or different scene as in the learning phase, they were no better than controls at linking these faces to the correct visual scene.

**2.2.3 UNSW House Test: Super-recognisers' memory and matching ability for non-face images.** In a final test, we examined whether super-recognisers were also better than controls in memory and matching tests involving images that did not contain people. To do this we tested them on the UNSW House Test, a version of the UNSW Face Test [33] that used images of houses instead of faces. Prior work has found that individual differences in face identity processing are somewhat related to abilities in processing other object types (e.g. [4,17,18,43]), suggesting that individual differences reflect graded expertise in visual processing rather than a sharp dissociation between more general visual processing and memory.

The UNSW House Test shares the same design as the UNSW Face Test and images of the same house are subject to the same image-level variability in camera angle, camera-to-subject-distance, lighting, etc. (see Fig 6 and Methods for further details on the test). Results of the UNSW House Test are shown in Fig 6. Super-recognisers (SR-Weak: M = 80.9%, SD = 15.2) were significantly more accurate than Students on this test (Students: M = 68.4%, *SD* = 13.6; $t(57.6) = 3.44$, $p = .001$, Cohen's $d = 0.86$), indicating they also have an advantage on non-face identification tasks, supporting the proposal that a domain-general visual matching ability may partly explain their superior face identification abilities.

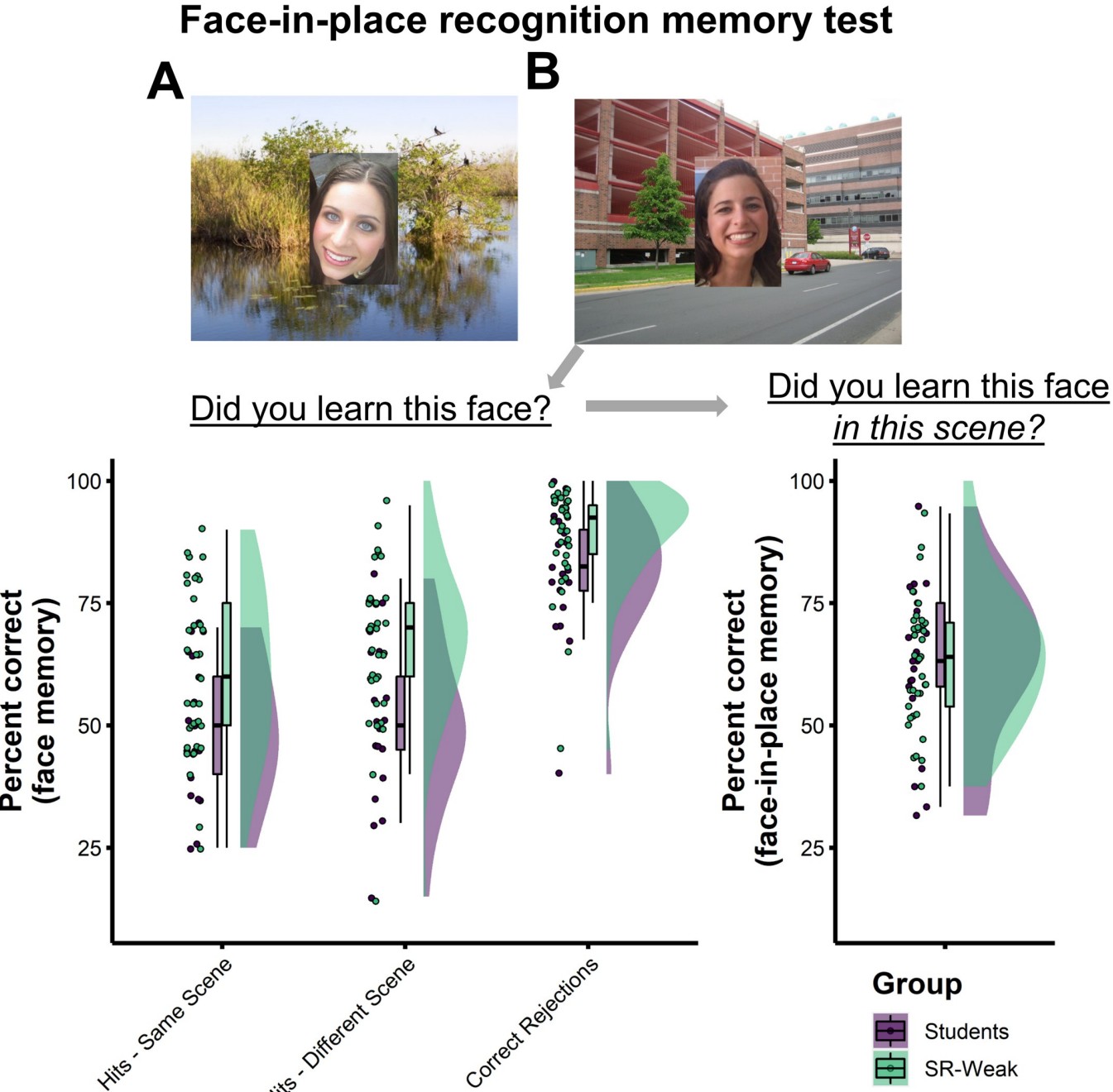

**Fig 5. Accuracy of super-recognisers and students on the face-in place recognition memory test.** (A) Participants learned faces embedded in scenes during the study phase. (B) Participants were then shown faces embedded in either same scenes or different scenes at test. (C) Participant accuracy shown separately for two different decisions made to face-scene images at test. Super-recognisers (SR-Weak) outperformed Student Controls when recognising previously encountered faces, both when shown in same and different visual scenes (Hits) and when correctly rejecting unseen faces (correct rejections). Super-recognisers were no better at deciding whether recognised faces were shown in same or different scenes (right plot).

### 2.3. Summary performance profiles show benefits of stricter criteria when selecting groups of super-recognisers

Fig 7 shows aggregated test results for super-recognisers on the face identification tests reported in the previous sections. Performance is expressed as mean z-scores for each super-

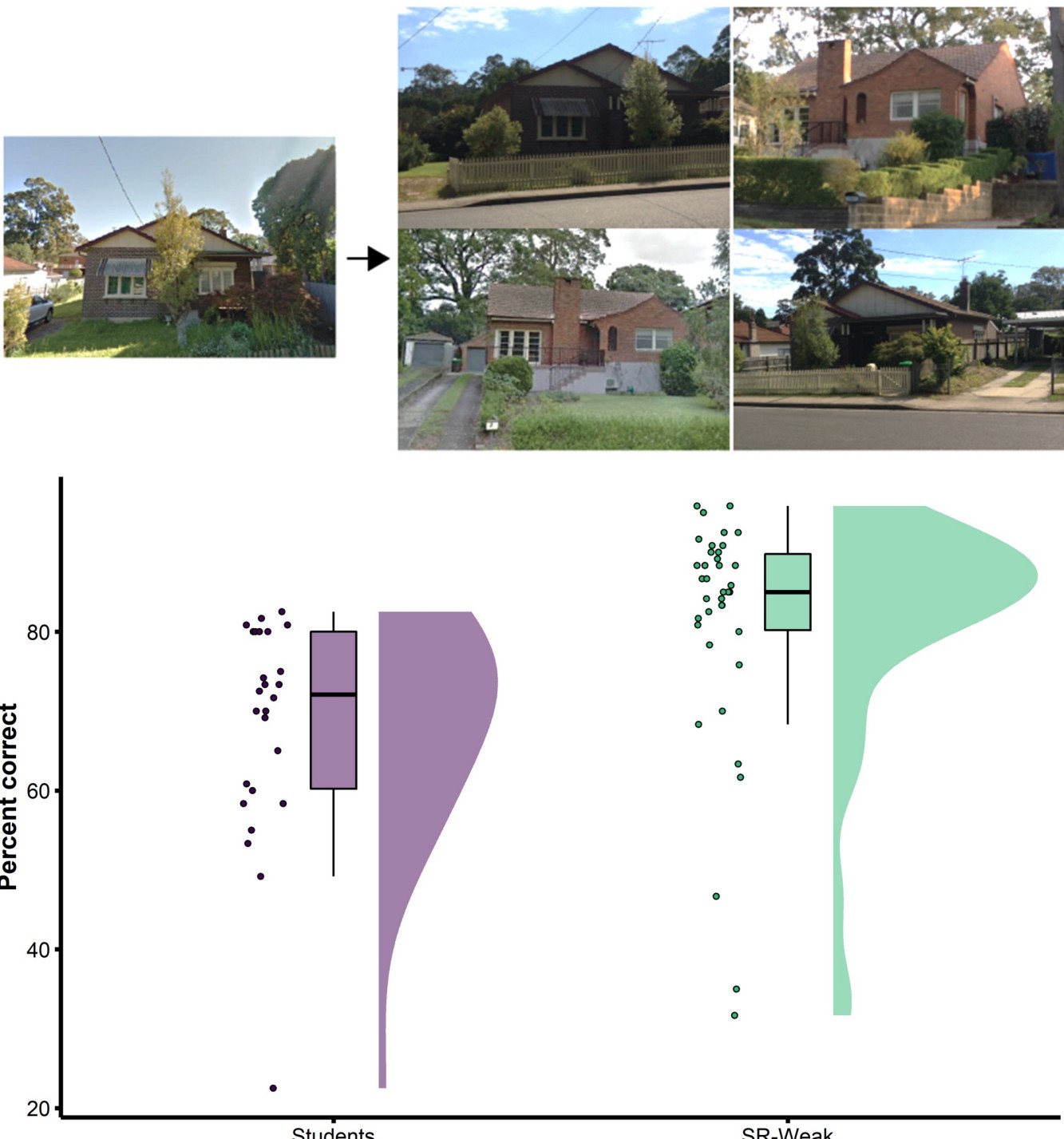

**Fig 6. Super-recogniser and student accuracy on the UNSW House Test.** The UNSW House Test examines whether super-recognisers' superior face matching and memory ability generalises to non-face objects like houses. Super-recognisers (SR-Weak) were more accurate on this test than Student Controls, suggesting that some of their superiority with faces generalises to the recognition of non-face objects.

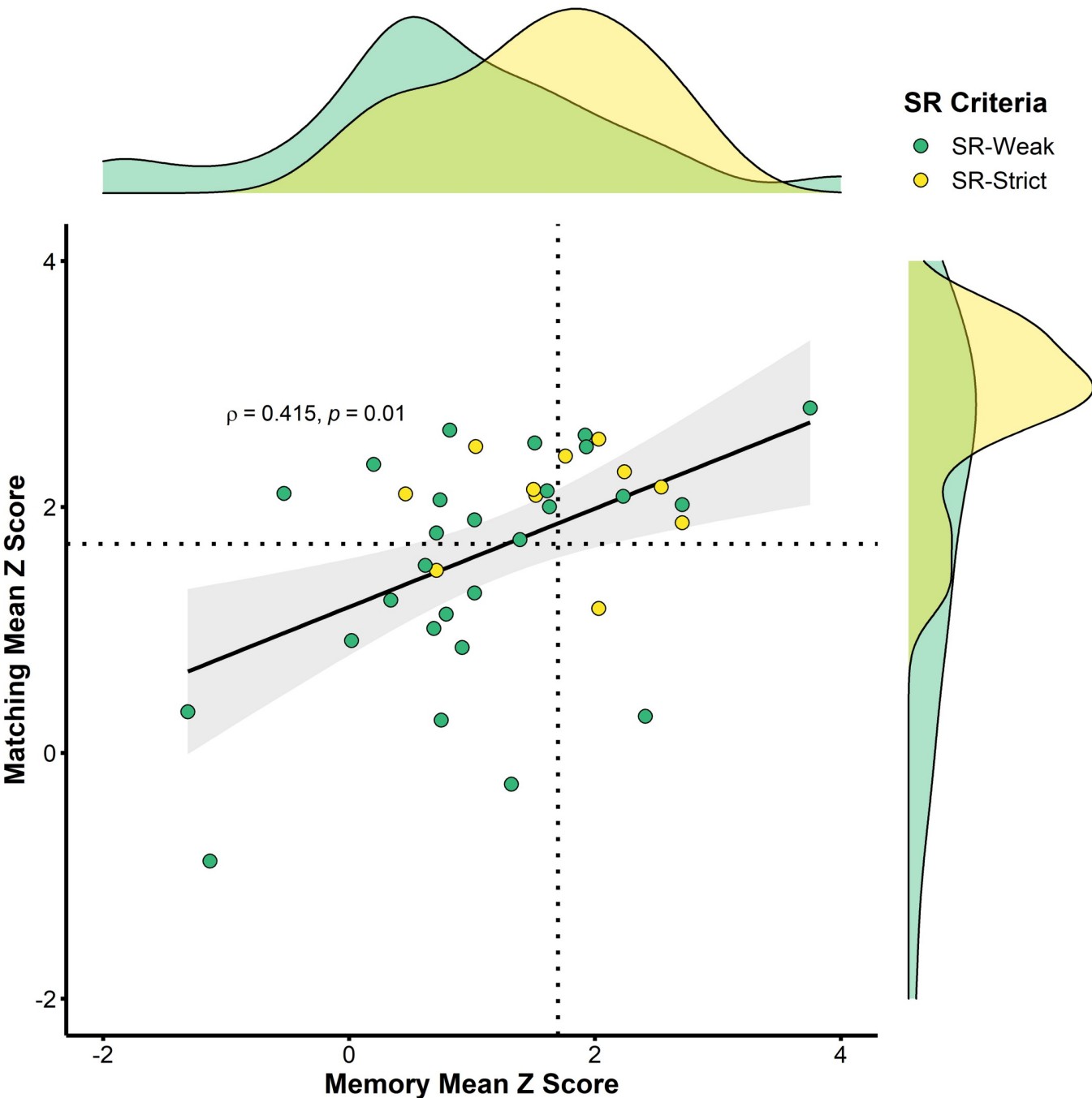

**Fig 7. Overall performance of super-recognisers across 7 follow-up face identification tests expressed as z-scores relative to control participant norms.**
Positive z-scores show that super-recognisers generally outperformed standard participant groups, but this was more consistent for the groups selected using strict performance criteria (yellow markers) than for those selected using weak criteria (green markers). Dotted lines represent the weak criteria as applied to the follow-up tests (1.7 SD), to illustrate the proportion of selected SRs that continued to meet SR criteria on follow-up testing. Details of this analysis are provided in the main text.

recogniser relative to control means and calculated separately for tests that required participants to memorise faces (2 tests, x-axis) and tests that required perceptual matching of face identity (5 tests, y-axis). For the purpose of this analysis, we omitted tests reported in this paper that: (i) were used as screening tests; (ii) were identification tests of non-face objects

(UNSW House Test, Body only trials in Face & Body test); or (iii) were inverted face matching tests (2 and 30s Inverted EFCT).

Next, we compared the effect of using a strict versus weak criteria to select super-recognisers. Across the follow-up lab-based tests, the strict criteria group had an average performance advantage over controls of $z = 1.68$ for memory and $z = 2.07$ for matching while the weak criteria group had an average performance advantage of $z = 1.23$ for memory and $z = 1.68$ for matching. In terms of accuracy, the strict group was 17% better than norms on memory tests and 28% better than norms on matching tests. This benefit was lower for the weak criteria group, who scored 12% better than norms for memory and 22% better for matching. Comparing performance between the strict and weak criteria groups, strict criteria super-recognisers were consistently more accurate than the weak criteria super-recognisers, scoring between 1–8% better on average for individual tests. The consistency of strict criteria group's outperformance of weak criteria was statistically reliable by a Wilcoxon sign test (strict criteria > weak criteria, $T_7 = 2.37$, $p = .018$).

Memory and matching performance were significantly positively correlated, *Spearman's rho* (37) = .415, $p = .01$. However, it is also clear from visual inspection of Fig 7 that individuals appear in all four quadrants of this plot: either passing the weak criteria (>1.7 SD) for follow-up tests on both memory and matching ($n = 10$, 26%), matching only ($n = 14$, 37%) or less commonly on memory only ($n = 2$, 5%). Further, there was a substantial number of participants that passed initial criteria based on online screening, but did not pass criteria on either memory or matching tasks after subsequent lab-based testing ($n = 12$, 32%).

Numerically, aggregate scores across all follow-up tests show superior performance for all but two of the super-recognisers. All of the super-recognisers selected on the basis of strict criteria showed substantial superiority over controls, and also higher accuracy relative to super-recognisers selected with weaker criteria. This verifies that the ability of super-recognisers persists beyond initial screening test performance. In addition, the clear variability in Fig 7 test scores points to the importance of using follow-up tests to verify individuals' ability (see also [31]) and applying stricter selection criteria using multiple tests to protect against selecting participants with average abilities.

## 3 Discussion

We tested face identification ability of over 1600 Australian police officers. From these, we selected a group of 38 for further lab-based testing that had achieved an average z-score of greater than +1.7 relative to normative accuracy on 3 standard tests. Prior studies have tested groups of super-recognisers that have been selected for roles in police forces based on face identity processing tests [20,21]. But this is the first full report, documented in a scientific journal, of a process used to select super-recognisers for a professional organisation, and verify their ability on a range of tests relating to operational practise.

Our results add to the cumulative evidence that super-recognisers can help improve face identification accuracy in applied settings [19,20,21,26,27]. As a group, super-recognisers consistently outperformed typical viewers across a range of lab-based face matching and memory tasks. On average they outperformed control participants by between 20% (weak criteria) and 25% (strict criteria) percentage points, which spans the same range as previous tests. This included high accuracy on a series of face matching tasks that were designed to reflect challenging decisions made in forensic settings, and this accuracy was equivalent to benchmark performance of forensic facial comparison expert groups.

Notably however, on these tests–which did not involve memory, and included tests with unlimited time to compare images–no super-recognisers, nor forensic examiners, achieved

perfect accuracy. Given the highly selective criteria and large pool of staff we chose super-recognisers from, this result suggests that the effective upper-limit of human's accuracy on challenging tasks–even those that do not require committing faces to memory–is substantially below 100%. Naturaly, memory demands led to lower accuracy, which should reject the claim that super-recognisers 'never forget a face'.

Individually, all but two super-recognisers in the weak criteria condition achieved accuracy that was numerically greater than controls on average. However, consistent with previous work, we observed substantial variation in the accuracy of individual super-recognisers in both memory and matching tests (e.g., [20,31]; see Fig 7) despite using multiple screening tests with diverse task demands, spanning both memory and matching ability. This underlines the difficulty of using current tests to identify individual super-recognisers that will, substantially and consistently, outperform average performers in tasks reflective of everyday police work. The potential of this approach is constrained by the reliability of tests [19,44] and the extent to which different types of face recognition ability tests converge on a unitary cognitive trait (i.e., 'convergent validity', see [8,18]). Since we conducted this work, tests reporting improved reliability have been published (see [18], Table 1), and researchers have begun working towards a standard framework for selecting super-recognisers using test batteries with more diverse task demands and measures [25,45]. It will be important in future work to verify whether these improvements can lead to quantitative gains in the performance of super-recognisers relative to the cohort we selected here.

For now, we have shown that a simple way to improve the effectiveness of selection is to apply stricter selection criteria. For our strict criteria group, where participants had achieved z-scores exceeding 1.7 standard deviations above normative accuracy on each of the 3 screening tests, we found that all super-recognisers consistently exceeded average performance on subsequent lab-based tests. Being highly selective can therefore mitigate against problems caused by limits on test reliability and validity.

Top performers on our screening tests showed high levels of accuracy on a battery of lab-based tests designed to capture some face identification tasks that might be performed in police work. Our battery of tests was not exhaustive, and did not include many important tasks performed in police work, for example monitoring video [45,46], live photo-ID checks [47] and longer-term recognition memory [29]. Ongoing evaluation of the effectiveness of the selection process we have documented here should therefore include testing of super-recognisers across more of the tasks they perform in their daily work.

Although ongoing testing beyond a single follow-up session was not included as part of this project, this type of evaluation will be important in future. This would provide further evidence of the effectiveness of the selection processes outlined here in improving face identification accuracy in police operations. It is also important to examine how SRs' performance changes over time with practice and training. In other forensic disciplines, fingerprint matching experts improve over time with training and experience [48] and interestingly, individual differences in ability prior to any training predicted their later accuracy. No such study has been conducted for forensic roles involving face processing, but we found comparable accuracy between trained facial examiners and super-recognisers, both here and in recent studies (e.g., [3,23]).

This suggests that super-recognisers could be selected for forensic examiner roles, but it remains unclear whether they would be amenable to forensic training (see 35, 50 for reviews). In separate work, we have also found evidence that SRs tend to make more high confidence errors relative to facial examiners [23,30]. This points to another important reason to test SR performance in detail after receiving training–to examine whether this overconfidence is attenuated by training and experience. High confidence errors can lead to serious negative

outcomes in criminal justice settings, for example wrongful convictions, and so this is an important question to address in future work.

Another aim of our study was to begin defining boundary conditions of SR's abilities. Previous work has shown that SRs tend to perform above average on visual processing and memory tasks that do not involve faces [20,23]. This is consistent with patterns of correlation between face and object processing in the broader ability spectrum [17,43,49]. Here we also found that SR groups outperformed controls on perceptual matching and memory tasks not involving faces, although individual SR performance was somewhat variable (c.f. [36]). The pattern of group difference appeared graded with respect to the type of object being processed, with face processing tasks showing the largest group difference, followed by bodies (Cohen's $d$ = 1.36) and buildings (Cohen's $d$ = 0.86). Similar graded patterns were found in a recent study where SRs showed advantage over controls in matching chimpanzee faces, which was reduced relative to their human face processing advantage, but greater than for fingerprint matching [23]. Together, this suggests that SRs possess a general aptitude for performing visual tasks, with their ability tuned towards face identity but affording residual benefit in similar visual processing tasks. Given pre-existing perceptual ability predicts later accuracy in fingerprint matching trainees [48], it would be of both applied and theoretical interest to know whether super-recognisers' skills predispose them to becoming experts in other object matching domains.

Finally, despite SRs' heightened ability to recognise faces, they had surprising difficulty in verifying the visual context that recognised faces had been presented in. Both super-recognisers and controls made approximately 40% errors in matching faces to the scenes they were presented in, demonstrating a clear boundary to SRs' expertise. Conversely, a previous study of memory 'athletes' with exceptional abilities to form face-name associations, found that they were no better at recognising faces [50]. As well as apparent practical implications, this may also provide insight into the nature of SR representations of person identity more broadly. It implies that highly sensitive perceptual representations of faces do not produce stronger connections with associated information. This dissociation is perhaps surprising given recent neuroscientific evidence that faces evoke more widely distributed brain activation in people with higher ability in face identification [51,52]. Understanding how super-recognisers link face memories to broader memory networks, including related person identity information, is therefore an important direction for future work.

In summary, we have shown that face identity processing abilities of super-recognisers selected from a large police force transfer well to a variety of lab-based tasks that reflect police work. Setting strict criteria can protect against selecting individuals that do not continue to show high levels of accuracy in subsequent testing. Some important qualifications remain. We have established boundary conditions that constrain the application of super-recognisers as a general solution for person identification tasks. Additionally, we have not evaluated performance of super-recognisers 'on the job', nor over a prolonged period. Because our lab-based tests did not capture the full diversity of operational tasks super-recognisers might be expected to perform, this type of work will be necessary in future to validate super-recogniser deployment in more complex real-world tasks.

# 4 Materials and methods

## 4.1 Ethics statement

This research was approved by the Deputy Vice Chancellor (Research) of the University of New South Wales on the advice of the Human Research Ethics Advisory Panel. All participants gave their informed consent either digitally or in writing. The faces of the individuals depicted

in the figures have given written informed consent (as outlined in PLOS consent form) to publish and use their faces for these purposes.

## 4.2 Participants

Approximately 17,000 staff NSW Police Force staff were invited to complete three screening tests that assessed their face identification ability (see below for details). At the end of phase 1, 2537 staff completed at least 1 screening test, and 1670 completed all 3. Top performing staff who completed all 3 screening tests were then invited to participate in additional lab-based testing in-person at the UNSW Sydney campus.

Average scores of police officers on the screening tests are shown along with normative data in Table 1. Police officers were equivalent to published normative scores for both the CFMT+ ($t(2042) = 0.22$, $p = .822$, Cohen's $d = 0.05$) and the UNSW Face Test ($t(2030) = 1.55$, $p = .122$, Cohen's $d = 0.12$) but were 8.2% more accurate on average than published normative scores for the GFMT ($t(2657) = 13.24$, $p < .001$, Cohen's $d = 0.85$). This may reflect the fact that GFMT is a self-paced test and so could be more susceptible to differences in participant motivation (but see [47]), but there also appears to be a general tendency for participant groups to score above normative in recent tests (see [16,53]).

To determine the effect of different super-recogniser selection thresholds, we recruited participants who satisfied either a strict (SR-Strict) or weak (SR-Weak) criteria. Individuals meeting SR-Strict criteria scored a minimum Z score of 1.7 SD on each screening test. Fifteen of 1651 participants (0.9%) satisfied these criteria and 11 participated in the lab tests ($M_{age} = 39.2$; 5 female, 6 male). Individuals meeting SR-weak criteria scored a minimum average Z score of 1.7 SD across the 3 screening tests. An additional 39 (2.4%) of the 1651 participants met SR-weak (not including those in the SR-Strict group) and 27 of those participated in the lab tests (Weak-only: $M_{age} = 37.3$; 10 female, 17 male). The ratio of female to male super-recognisers was proportionate to the composition of the NSW Police Force that completed all 3 the screening tests (female = 39%, male = 60%, other or prefer not to say 1%). This suggests that the selection process resulted in a gender balance that reflected the initial test cohort (for female: male ratio of faces used in our tests see S1 Appendix).

## 4.3 Screening tests

All staff members of the NSW Police Force were invited to participate in the super-recogniser screening research program via email and a notice on their intranet home page. Clicking a link in the advertisement directed participants to a central website where they could provide informed consent to participate and access 3 standardised tests of face identification ability. Participants were instructed that they could complete the tests in any order and in separate sessions, but that after commencing a test they must complete it in a single session. Participants were not given any feedback or told their scores on each test until the end of the screening test period, which concluded after a month. The three screening tests were as follows.

**Cambridge Face Memory Test (CFMT+).** The CFMT+ (7) is a standardised test of unfamiliar face memory ability. Participants' learn and recognize unfamiliar faces in a three-alternative-forced-choice test. The test contains four stages that successively increase the difficulty of the task by introducing either untrained views and lighting conditions of the images or by adding visual noise to the images. Accuracy is computed from the percentage of trials with correct identifications out of 102 and normative data for this test was sourced from (7).

**Glasgow Face Matching Test (GFMT).** In the short version of the GFMT [4], participants match 40 face pairs (20 match, 20 nonmatch) by deciding whether they show the same person or two different people. Images of the same person were captured on the same day, just

minutes apart, but with different cameras. Accuracy is computed from the percentage of trials with correct decisions out of 40. Normative data for this test was sourced from (4).

**UNSW Face Test.** The UNSW Face Test [33] is a screening test calibrated to be very difficult and so is suited to screening for identify super-recognisers. It consists of two tasks completed in a fixed order: a recognition memory task and a match-to-sample sorting task. For further details of this test see S1 Appendix. Accuracy is computed from the percentage of correct recognition and sort decisions out of 120. Normative data for this test was sourced from [33].

### 4.4 Comparing police super-recognisers to facial examiners on challenging face identification tests in the lab verifies their superior ability

Further details of the tests described below, performance measures and extended reporting of analysis are available in S1 Appendix. All follow up comparisons were conducted using Welch's t-test as it is generally best when there is a difference between the variance of two populations and also when their sample sizes are unequal.

**4.4.1 Facial recognition candidate list test.** The facial recognition candidate list test models the daily work of passport issuance officers using facial recognition technology to screen for passport fraud and has been used in prior work to test the accuracy of facial examiners ([22]: Experiment 2). The test requires participants to identify whether a target face appears in a gallery of eight faces and to select the correct target when present. Target faces appear in the candidate list on 50% of trials in a random position.

**4.4.2 Selfie-to-passport test.** The Selfie-to-passport test was designed to measure accuracy on images that were the most difficult for facial recognition algorithms to classify. This test contained a total of 53 trials– 19 match and 34 nonmatch trials consisting of various ethnicities representative of the Australian population. On each trial, participants were presented with two face images simultaneously–one 'selfie' image and one passport image.

**4.4.3 Expertise in facial comparison test (EFCT).** The EFCT is a pairwise unfamiliar face matching test that has previously been used to test an international group of forensic facial examiners (see [32]). The EFCT involves completing the same two subtests twice: first with images shown on screen for 2-seconds each, and later with images shown on screen for 30-seconds each. All subtests contain 84 face pairs (half same-identity, half different identity) and one is completed with faces shown upright, and the other with faces inverted.

### 4.5 Establishing the boundaries of super-recognisers' superior abilities

Further details of the tests described below, performance measures and extended reporting of analysis are available in S1 Appendix.

**4.5.1 Face and body matching test.** The Face and body matching test examined participants' ability to identify people based on only their face or only their body. This test consisted of 40 image pairs, 20 that were of the same person (match) and 20 of two different people (nonmatch). These images were sourced from the Person Identification Challenge Test ss32] but were cropped here to show only the face or only the body (see Fig 3 for example), resulting in a total of 80 image pairs (40 face only, 40 body only).

Forty-six undergraduate students were recruited as control participants for this test in exchange for course credit. This sample consisted of 29 females and 17 males aged between 18 and 24 (mean age = 19.0, SD = 1.3).

**4.5.2 Photoboard recognition memory test.** We designed a bespoke test of face memory performance to model a scenario where police officers are asked to memorise faces studied on 'photoboards', and to later recognise them (see Fig 4 for example).

This test was separated into 2 blocks, with a study phase, filler task and test phase in each block. The study phase contained 16 identities for the participants to remember. Each identity was shown in 1 single image or 8 different images for a total of 10 seconds. Following the study phase, participants were asked to complete a filler word-search task for 5 minutes. Then in the test phase, participants saw 2-second videos of 16 studied and 16 unstudied faces that showed the person either slowly turning towards the camera (90–0˚) or away from camera (0–90˚). For each video, participants responded whether or not they had studied that person previously. Half the videos showed previously studied faces captured in a different location to the photos shown in the learning phase, and the other half showed novel identities. This process was repeated again with a new set of 16 faces, and another word-search filler task. The faces assigned to each condition were counterbalanced between participants.

Twenty-three undergraduate students were recruited as control participants for this test in exchange for course credit. This sample consisted of 14 females (13 males) aged between 18 and 25 (mean age = 19.1, SD = 1.5).

**4.5.3 Face-in-place recognition memory test.** The face-in-place recognition memory test was developed to examine the extent to which people remember or use the visual context in which they first encountered a face during a subsequent recognition test. In a study phase, participants were presented with a sequence of 20 face images placed in the center of a visual scene (see Fig 5 for an example).

This test was composed of a study phase, and a test phase. In each trial of the study phase, participants saw a face image placed in the center of a visual scene (see Fig 5 for an example). Participants were presented with a total of 20 images, each showing a different person in a different scene. Images were shown in a randomized order for 10 seconds each. The test phase consisted of 80 trials split into the following conditions: 20 of the studied faces were shown twice in the test phase but in different images, once on the same background as the study phase and once on a different scene to the study phase (total of 40 match trials, 20 on the same scene, 20 on a different scene). Participants were also shown 40 foil faces randomly intermixed with the studied faces, with the foil faces all shown on one of the scenes shown with studied faces (total of 40 nonmatch trials).

In each trial of the test phase, participants were shown a face in a scene, and asked to decide if they recognised the face from the study phase (face recognition accuracy). When participants said they recognised the face, they were then asked if they had seen that face on the same scene (face-in-place recognition).

Twenty-one undergraduate students were recruited as control participants for this test in exchange for course credit. This sample consisted of 13 females and 8 males aged between 18 and 21 (mean age = 19.0, SD = 1.0).

**4.5.4 UNSW House Test.** The UNSW House Test was developed to test participants' matching accuracy for non-face objects using the same format as the UNSW Face Test. This test therefore allows us to measure the extent to which performance on the UNSW Face Test is related to face-specific processing, compared to non-face object processing. The UNSW House Test contains a Recognition Memory Task and a Match-to-Sample Sorting task, completed in the same order for each participant. The images of the same house are subject to the same image-level variability in camera angle, camera-to-subject-distance, lighting etc. Accuracy is computed from the percentage of correct recognition and sort decisions out of 120. Fig 6 shows illustrative examples of images used in this test and further details are available in S1 Appendix.

Twenty-six undergraduate students were recruited as control participants for this test in exchange for course credit. This sample consisted of 17 females and 9 males aged between 18 and 24 (mean age = 19.0, SD = 1.5).

## Supporting information

**S1 Appendix. Additional data and analysis.**
(DOCX)

## Acknowledgments

Thank you to Daniel Guilbert, Gabrielle Killen, Bojana Popovic, Melissa Bebbington, Alicia Jung, William Hoang, and Victor Varela for assistance with data collection, and to Daniel Guilbert for assistance developing the UNSW House Test.

## Author Contributions

**Conceptualization:** Alice Towler, David White.

**Data curation:** James D. Dunn.

**Formal analysis:** James D. Dunn.

**Funding acquisition:** Alice Towler, Richard I. Kemp, David White.

**Investigation:** James D. Dunn, Richard I. Kemp, David White.

**Methodology:** James D. Dunn, Alice Towler, David White.

**Project administration:** James D. Dunn, Alice Towler.

**Supervision:** Richard I. Kemp, David White.

**Visualization:** James D. Dunn.

**Writing – original draft:** James D. Dunn, David White.

**Writing – review & editing:** James D. Dunn, Alice Towler, Richard I. Kemp, David White.

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
