## [Decision Letter · Decision Letter 0]

26 Dec 2022

PONE-D-22-32244Selecting police super-recognisersPLOS ONE

Dear Dr. White,

Thank you for submitting your manuscript to PLOS ONE. After careful consideration, we feel that it has merit but does not fully meet PLOS ONE’s publication criteria as it currently stands. Therefore, we invite you to submit a revised version of the manuscript that addresses the points raised during the review process.

We look forward to receiving your revised manuscript.

Kind regards,

Rocco Palumbo

Academic Editor

PLOS ONE

Journal Requirements:

4. We note that Figure 2 includes an image of a participant in the study. 

Reviewers' comments:

Reviewer's Responses to Questions

**Comments to the Author**

1. Is the manuscript technically sound, and do the data support the conclusions?

Reviewer #1: Yes

Reviewer #2: Yes

2. Has the statistical analysis been performed appropriately and rigorously? 

Reviewer #1: I Don't Know

Reviewer #2: Yes

3. Have the authors made all data underlying the findings in their manuscript fully available?

Reviewer #1: Yes

Reviewer #2: Yes

4. Is the manuscript presented in an intelligible fashion and written in standard English?

Reviewer #1: Yes

Reviewer #2: Yes

5. Review Comments to the Author

Reviewer #1: This is a very interesting paper describing selection of super-recognisers from a large organisation. While most of the work in this area tends to be descriptive or small-group lab-based, the authors have offered a major innovation here. This has the potential to become a very influential paper, and I recommend publication. Its main contributions are (i) description of organisational-level SR-selection; and (ii) the very interesting finding that SR performance does not generalise well to other tasks that seem related.

I have only two main comments to make.

First, I am a little confused about the direct statistical comparison of SR-Strict and SR-Weak groups (for example line 295 et passim). Am I right to think that the Strict group is a subset of the Weak group? It is hard to work this out, because of the non-integer df values (presumably due to a statistical correction?). If one of the SR groups is a subset of the other, then it is not straightforward to see how these can be compared with a simple t-test. Of course, it is easy to see how each SR group can be compared with other groups (e.g. line 291 et passim), and the size of any difference can be discussed. Those are straightforward between-subjects comparisons, but the comparison of overlapping groups is not like this. Perhaps I am mistaken here, and the authors can clarify the procedure.

Second, I was not convinced that the summary beginning on line 212 was very helpful. The problem is that the reader is presented with aggregate data before knowing what the constituent tests were. This would be fine if it were straightforward, but even the aggregation comes with qualifications (i.e. about excluded tests – line 220), and these qualifications do not really make sense until one has read the detailed material that follows. In fact, I also found presentation of Fig 1 in this section to be a bit unhelpful here. It means that the very first visual presentation of data encourages the reader to consider an association between memory and matching that is not at all central to the purpose of the paper. For me, figure 2 would be a much better place to start. So, I would suggest that the authors consider putting the material in the section starting line 212 towards the end of their results section, not at the beginning. Note that this will not introduce narrative tension, as the results have already been summarised in an earlier section (line 160), which is good. I emphasise that this is just a suggestion for readability – not a condition for acceptance.

Finally, I noticed a typo on line 168: “consistently higher” missing verb

Reviewer #2: Review of the manuscript entitled “Selecting police super-recognisers” by White, Dunn, Towler and Kemp

This is a very interesting study in which the Authors exploited a dozen of validated paradigms to investigate facial perception and memory in a group of Australian police “super-recognisers” (SRs). The sample was divided into two subgroups: weak and strict SRs and their performance was compared to normative data or, when these data were not available, with students’ performance. The study is well-conceived and the manuscript well-written. The set of paradigms explored an interesting set of different skills in this specific population, and I am happy to endorse the publication of this work, after suggesting some minor hints which could help in further improve the clarity of the study:

1. The final sample is not so wide to allow the authors to include Sex as a between-subjects factor, but some more details on the gender of the stimuli would be useful for each paradigm. Is there a possible gender difference (both sample and stimuli) in such a domain? This topic should be at least named in the manuscript.

2. In the Introduction, brain and physiology difference in face recognition and face manipulation are named – I suggest to insert a further citation on hemispheric asymmetries for face recognition and spatial manipulation: https://www.ncbi.nlm.nih.gov/pmc/articles/PMC4671171/

3. To facilitate the reading of the manuscript, the set of paradigm could be numbered (from 1 to 12, including selection criteria).

4. Page 12 starts with a significant 3-way ANOVA, but then two separate 2-way ANOVA are described. Since the 3-way interaction is significant, I strongly suggest reporting the overall interaction without splitting into two different analyses (which makes it impossible to see the real interaction among the three factors).

5. Line 344: is “Exposure duration” the same factor named “Study duration” some lines above? Please, check for consistency in variable names.

6. Line 353: the citation is not written according to the journal standard (numbers).

7. Page 14, line 413: the ANOVA interaction was significant: please, insert post-hoc comparisons before other (main effects) significant results.

6. PLOS authors have the option to publish the peer review history of their article (what does this mean?). If published, this will include your full peer review and any attached files.

Reviewer #1: No

Reviewer #2: No

---

## [Author Response · Author response to Decision Letter 0]

19 Jan 2023

James D. Dunn, Alice Towler, Richard I. Kemp, and David White

Author response to editor and reviewer’s manuscript PONE-D-22-32244

Selecting police super-recognisers

Author reply is denoted by ** and bold text. All page and line references refer to the tracked changes version.

Journal Requirements:

**Manuscript formatting has been updated to adhere to style requirements.

**Ethics statement has been added under the Method section.

**Caption for supporting information file have been added to the end of the manuscript before the results section.

4. We note that Figure 2 includes an image of a participant in the study. 

**We confirm that consent has been given from the individual’s pictures to use their image for publication. Note that we have switched the images in Fig 5 for images of a different person who has signed informed consent. The ethics statement now includes a statement about the faces in the figures: “The faces of the individuals depicted in the figures have given written informed consent (as outlined in PLOS consent form) to publish and use their faces for these purposes.”

**Reference list has been reviewed and is correct.

Comments to the Author

Reviewer #1

This is a very interesting paper describing selection of super-recognisers from a large organisation. While most of the work in this area tends to be descriptive or small-group lab-based, the authors have offered a major innovation here. This has the potential to become a very influential paper, and I recommend publication. Its main contributions are (i) description of organisational-level SR-selection; and (ii) the very interesting finding that SR performance does not generalise well to other tasks that seem related.

I have only two main comments to make.

First, I am a little confused about the direct statistical comparison of SR-Strict and SR-Weak groups (for example line 295 et passim). Am I right to think that the Strict group is a subset of the Weak group? It is hard to work this out, because of the non-integer df values (presumably due to a statistical correction?). If one of the SR groups is a subset of the other, then it is not straightforward to see how these can be compared with a simple t-test. Of course, it is easy to see how each SR group can be compared with other groups (e.g. line 291 et passim), and the size of any difference can be discussed. Those are straightforward between-subjects comparisons, but the comparison of overlapping groups is not like this. Perhaps I am mistaken here, and the authors can clarify the procedure.

** The reviewer is correct that statistical comparison of the strict V weak criteria SR groups is complicated by the fact that the SR-strict group is a subset of the SR-weak group. On the other hand, comparing the relative average accuracy of these groups is important practically, because it quantifies the accuracy that would be expected given two selection criteria. As a solution, we have removed direct t-test comparisons in favour of reporting the effect size of the differences in individual tests, and we refer the reader to the summary analysis at the end for a formal analysis (which we have now moved to the end following R1’s next comment). 

Regarding the non-integer df values, we used a Welch t-test for all the planned comparisons (this is now noted on pg 26 ln 715). Welch's t-test also known as unequal variances t-test is generally applied when there is a difference between the variance of two populations and also when their sample sizes are unequal. We now note this near line 933.

Second, I was not convinced that the summary beginning on line 212 was very helpful. The problem is that the reader is presented with aggregate data before knowing what the constituent tests were. This would be fine if it were straightforward, but even the aggregation comes with qualifications (i.e. about excluded tests – line 220), and these qualifications do not really make sense until one has read the detailed material that follows. In fact, I also found presentation of Fig 1 in this section to be a bit unhelpful here. It means that the very first visual presentation of data encourages the reader to consider an association between memory and matching that is not at all central to the purpose of the paper. For me, figure 2 would be a much better place to start. So, I would suggest that the authors consider putting the material in the section starting line 212 towards the end of their results section, not at the beginning. Note that this will not introduce narrative tension, as the results have already been summarised in an earlier section (line 160), which is good. I emphasise that this is just a suggestion for readability – not a condition for acceptance.

** On reflection we agree with the reviewer and thank them for the suggestion, the summary analysis is now reported at the end of the Methods and Results section. 

Finally, I noticed a typo on line 168: “consistently higher” missing verb

**This error has been corrected.

Reviewer #2

This is a very interesting study in which the Authors exploited a dozen of validated paradigms to investigate facial perception and memory in a group of Australian police “super-recognisers” (SRs). The sample was divided into two subgroups: weak and strict SRs and their performance was compared to normative data or, when these data were not available, with students’ performance. The study is well-conceived and the manuscript well-written. The set of paradigms explored an interesting set of different skills in this specific population, and I am happy to endorse the publication of this work, after suggesting some minor hints which could help in further improve the clarity of the study:

1. The final sample is not so wide to allow the authors to include Sex as a between-subjects factor, but some more details on the gender of the stimuli would be useful for each paradigm. Is there a possible gender difference (both sample and stimuli) in such a domain? This topic should be at least named in the manuscript.

** All tests had roughly equal numbers of male and female faces (varying between 40:60 – 60:40 in ratio) except for the CFMT+ which only contains male faces. Below is a table showing the female:male ratio in each test which shows a consistently even ratio across tests. 

Test % of female faces in test

GFMT 40:60

UNSW Face Test 50:50

Facial recognition candidate list test 50:50

Selfie-to-passport test 60:40

EFCT Upright 54:46

EFCT Inverted 61:39

Face and body matching test 40:60

Photoboard recognition memory test 50:50

Face-in-place recognition memory test 50:50

Gender differences in face identity processing individual difference tests tend to be very small (e.g. S1 Appendix in Dunn et al., 2020), and not consistently observed across studies either in the presence of a difference or the direction. 

To test if there was any evidence that our selection process was biased towards selecting one gender over another, we compared the female:male ratio for the selected super-recognisers (39:61) to the female:male ratio of the NSW Police Force officers that completed all 3 the screening tests (40:60). This suggests that the selection process resulted in a gender balance that reflected the initial test cohort.

2. In the Introduction, brain and physiology difference in face recognition and face manipulation are named – I suggest to insert a further citation on hemispheric asymmetries for face recognition and spatial manipulation: https://www.ncbi.nlm.nih.gov/pmc/articles/PMC4671171/

** The reference to physiology here was relating to associations between physiology and face identity processing ability. While we do agree that relationship between hemispheric asymmetries and face identity processing ability would be an interesting research topic, as the study cited here does not appear to directly test this association, we have opted not to cite in our introduction.

3. To facilitate the reading of the manuscript, the set of paradigm could be numbered (from 1 to 12, including selection criteria).

**We considered this as it may have made cross-referencing easier, but after making the necessary edits we felt that it was not helpful and perhaps distracting. So we have retained the headings from the previous version of the ms. 

4. Page 12 starts with a significant 3-way ANOVA, but then two separate 2-way ANOVA are described. Since the 3-way interaction is significant, I strongly suggest reporting the overall interaction without splitting into two different analyses (which makes it impossible to see the real interaction among the three factors).

** The reviewer is correct that our ‘follow-up’ analysis did not address the nature of the 3-way interaction. However, we believe that the original presentation of the results somewhat mischaracterised our approach to analysis. In fact, we had intended a priori to analyse the effects of Study Duration and the Inversion Effect separately and this was not in fact a follow-up of the 3-way ANOVA. 

We have revised this results section (from line 378) to make this clearer. We now report the full 3-factor ANOVA model in the Supplementary Materials with a Supplementary Figure to aid interpretation of the full ANOVA model. We believe the results of the 3-way ANOVA are consistent with the analysis reported in the main paper but that computing face inversion effects for individual participants aids interpretation of the pattern of results. 

5. Line 344: is “Exposure duration” the same factor named “Study duration” some lines above? Please, check for consistency in variable names.

**All instances of ‘Exposure duration’ have been corrected to Study duration.

6. Line 353: the citation is not written according to the journal standard (numbers).

**This reference has been corrected.

7. Page 14, line 413: the ANOVA interaction was significant: please, insert post-hoc comparisons before other (main effects) significant results.

**The tests that are reported in this section were already reporting simple main effects. Main effects are reported in supplementary materials as these were qualified by the sig interaction. We have added a reference to the supplementary materials here as it was missing.

---

## [Decision Letter · Decision Letter 1]

20 Feb 2023

PONE-D-22-32244R1Selecting police super-recognisersPLOS ONE

Dear Dr. White,

Thank you for submitting your manuscript to PLOS ONE. After careful consideration, we feel that it has merit but does not fully meet PLOS ONE’s publication criteria as it currently stands. Therefore, we invite you to submit a revised version of the manuscript that addresses the points raised during the review process.

We look forward to receiving your revised manuscript.

Kind regards,

Rocco Palumbo

Academic Editor

PLOS ONE

Journal Requirements:

Reviewers' comments:

Reviewer's Responses to Questions

**Comments to the Author**

1. If the authors have adequately addressed your comments raised in a previous round of review and you feel that this manuscript is now acceptable for publication, you may indicate that here to bypass the “Comments to the Author” section, enter your conflict of interest statement in the “Confidential to Editor” section, and submit your "Accept" recommendation.

Reviewer #1: All comments have been addressed

Reviewer #2: (No Response)

2. Is the manuscript technically sound, and do the data support the conclusions?

Reviewer #1: (No Response)

Reviewer #2: Yes

3. Has the statistical analysis been performed appropriately and rigorously? 

Reviewer #1: (No Response)

Reviewer #2: Yes

4. Have the authors made all data underlying the findings in their manuscript fully available?

Reviewer #1: (No Response)

Reviewer #2: Yes

5. Is the manuscript presented in an intelligible fashion and written in standard English?

Reviewer #1: (No Response)

Reviewer #2: Yes

6. Review Comments to the Author

Reviewer #1: (No Response)

Reviewer #2: Review 2 of the manuscript entitled “Selecting police super-recognisers” by White, Dunn, Towler and Kemp

Some of my previous comments have been not considered by the Authors, I would like to suggest a further effort in considering them. Specifically, from my previous points:

Previous point 1. Sex difference should be at least named in the manuscript:

- A personal reply has been added in the letter, but Point 1 is anyway omitted in the manuscript. I suggest inserting this point also in the discussion, also by inserting the explanations provided in the letter.

Previous point 2. A further citation on hemispheric asymmetries for face recognition and spatial manipulation: https://www.ncbi.nlm.nih.gov/pmc/articles/PMC4671171/

- I understand that hemispheric asymmetries are not crucial for the present study, but I believe the suggested citation can be interesting for a potential reader of this manuscript. Obviously, the Authors are free to decide whether this reference is of interest – my personal opinion is favourable to this option.

Previous point 3. To facilitate the reading of the manuscript, the set of paradigms could be numbered:

- I still believe that the numbering would make it much easier to follow the manuscript.

7. PLOS authors have the option to publish the peer review history of their article (what does this mean?). If published, this will include your full peer review and any attached files.

Reviewer #1: No

Reviewer #2: No

---

## [Author Response · Author response to Decision Letter 1]

6 Mar 2023

Author reply is denoted by **. All page and line references refer to the tracked changes version.

Reviewer 1

Reviewer 1 did not have additional comments on this revision. 

Reviewer 2

Some of my previous comments have been not considered by the Authors, I would like to suggest a further effort in considering them. Specifically, from my previous points:

Previous point 1. Sex difference should be at least named in the manuscript:

- A personal reply has been added in the letter, but Point 1 is anyway omitted in the manuscript. I suggest inserting this point also in the discussion, also by inserting the explanations provided in the letter.

**We have now added an additional sentence describing these sex differences in the manuscript (Participant’s line 679) and included female:male ratios of faces in the tests to Supplementary Materials (Supplementary Table 2). 

Previous point 2. A further citation on hemispheric asymmetries for face recognition and spatial manipulation: https://www.ncbi.nlm.nih.gov/pmc/articles/PMC4671171/

- I understand that hemispheric asymmetries are not crucial for the present study, but I believe the suggested citation can be interesting for a potential reader of this manuscript. Obviously, the Authors are free to decide whether this reference is of interest – my personal opinion is favourable to this option.

**We continue to believe that this paper is not sufficiently relevant to our study to warrant inclusion in the reference list because it is not about individual differences in face recognition ability. 

Previous point 3. To facilitate the reading of the manuscript, the set of paradigms could be numbered:

- I still believe that the numbering would make it much easier to follow the manuscript.

**We have now numbered the subsections of Method & Results and refer to these in the final paragraph of the introduction.

---

## [Editor Report · Decision Letter 2]

14 Mar 2023

Selecting police super-recognisers

PONE-D-22-32244R2

Dear Dr. White,

We’re pleased to inform you that your manuscript has been judged scientifically suitable for publication and will be formally accepted for publication once it meets all outstanding technical requirements.

Kind regards,

Rocco Palumbo

Academic Editor

PLOS ONE
---

## [Editor Report · Acceptance letter]

23 Mar 2023

PONE-D-22-32244R2 

Selecting police super-recognisers 

Dear Dr. White:

I'm pleased to inform you that your manuscript has been deemed suitable for publication in PLOS ONE. Congratulations! Your manuscript is now with our production department. 

Kind regards, 

on behalf of

Dr. Rocco Palumbo 

Academic Editor

PLOS ONE